# Molecular and Metagenomic Analyses Reveal High Prevalence and Complexity of Viral Infections in French-American Hybrids and North American Grapes

**DOI:** 10.3390/v15091949

**Published:** 2023-09-19

**Authors:** Huogen Xiao, Baozhong Meng

**Affiliations:** 1Department of Molecular and Cellular Biology, University of Guelph, 50 Stone Road East, Guelph, ON N1G 2W1, Canada; 2Agriculture and Agri-Food Canada, Summerland Research and Development Centre, 4200 Highway 97, Summerland, BC V0H 1Z0, Canada

**Keywords:** RT-PCR, virus survey, RNA-Seq, grapevine, French-American hybrid grapes, non-*vinifera* grapes, Vidal, virome, GLRaV-3, GRBV, GPGV, TSV

## Abstract

French-American hybrids and North American grape species play a significant role in Canada’s grape and wine industry. Unfortunately, the occurrence of viruses and viral diseases among these locally important non-*vinifera* grapes remains understudied. We report here the results from a large-scale survey to assess the prevalence of 14 viruses among 533 composite samples representing 2665 vines from seven French-American hybrid wine grape cultivars, two North American juice grape cultivars (Concord and Niagara), and the table grape cultivar Sovereign coronation. Based on reverse transcription polymerase chain reaction (RT-PCR) assays, ten viruses were detected. Grapevine rupestris stem pitting-associated virus, grapevine leafroll-associated virus 3, grapevine Pinot gris virus and grapevine red blotch virus were detected with the highest frequency. As expected, mixed infections were common; 62% of the samples contained two or more viruses. Overall, hybrid wine grapes were infected with more viruses and a higher prevalence of individual viruses than juice and table grapes. To validate these findings and to refine the virome of these non-European grapes, high-throughput sequencing (HTS) analyses of five composite samples representing each category of grapevine cultivars was performed. Results from HTS agreed with those from RT-PCR. Importantly, Vidal, a widely grown white-wine grape with international recognition due to its use in the award-winning icewine, is host to 14 viruses, four of which comprise multiple and distinct genetic variants. This comprehensive survey represents the most extensive examination of viruses among French-American hybrids and North American grapes to date.

## 1. Introduction

Grapevine (*Vitis* spp.) is economically one of the most important and widely grown fruit crops worldwide. Global grape production in 2020 reached 77 million metric tons (Food and Agriculture Organization). Grapes are used mainly for wine production, but also as table grapes and for raisins, juice, vinegar, and other products [1]. Most grape cultivars are of *V. vinifera*, the common grapevine native to the Mediterranean and Central Asia. In the mid-1800s, the North American pests phylloxera and powdery mildew were accidentally brought to the Mediterranean region and central Europe, destroying most vineyards there. French breeders responded by developing new varieties using wild grapevine species native to North America that are resistant to phylloxera and powdery mildew [2]. Thereafter, two further waves of breeding efforts led to the modern hybrid grapes, which are still commonly grown in North America [1,2]. Furthermore, minor amounts of fresh fruit and wine come from North American species such as *V. labrusca* (the fox grape), *V. mustangensis* (the mustang grape), *V. rotundifolia* (the muscadines), and *V. riparia* (the riverbank grape), together with *V. amurensis* (the most important grape species from Asia). For instance, several cultivars of juice grapes were developed based on *V. labrusca*, including Concord and Niagara. These locally important *V. labrusca* grapes were widely grown in the Great Lakes of USA and Canada [1], although acreage has been dropping rapidly in recent decades.

Viruses are detrimental pathogens and are responsible for significant economic losses to grape and wine production worldwide. At present, grapevines are known to be infected by more than 80 viruses that belong to 29 genera and 17 families [3], making grapevine the plant crop infected with the most numerous viruses. Viruses involved in disease complexes such as grapevine leafroll (GLRD), infectious degeneration and decline, rugose wood (RW) and the more recently identified grapevine red blotch (GRB) rank at the top in terms of economic losses. Among the six viruses associated with GLRD, grapevine leafroll-associated virus 1 (GLRaV-1), GLRaV-2, and GLRaV-3 have a global presence with high prevalence and have been widely detected in Ontario [4]. Another complex of diseases, RW, is responsible for the decline of newly established vineyards, leading to total crop loss several years after infection. Though the etiology of RW diseases is yet to be resolved, grapevine rupestris stem pitting-associated virus (GRSPaV) and several viruses of the genus *Vitivirus*, including grapevine virus A (GVA) and GVB, are likely involved. Infectious degeneration and decline are due to infection by multiple viruses of the genus *Nepovirus* (family *Secoviridae*), including grapevine fanleaf virus (GFLV), arabis mosaic virus (ArMV), tomato ringspot virus (ToRSV) among others. Moreover, two new diseases have been recognized recently: grapevine red blotch caused by grapevine red blotch virus (GRBV), a single-stranded DNA virus of the family *Geminiviridae* [5], and grapevine leaf mottling and deformation (GLMD) with grapevine Pinot gris virus (GPGV) as the putative causal agent [6]. The wide application of high-throughput sequencing (HTS) technology has enabled the discovery of numerous new viruses from grapevine, including several viruses of the genus *Vitivirus* (family *Betaflexiviridae*) [7,8,9,10,11]. However, the pathological properties and economic impact of these recently identified viruses in grapevine is unknown.

Viral diseases have impeded sustained production and profitability of the Ontario grape/wine industry. Understanding the types of viruses, along with their prevalence and disease severity, is the first step in the battle against viruses and the diseases they cause. To this end, we completed province-wide surveys in 2015–2016 to understand the situation of viruses in *V. vinifera* wine grapes in Ontario [4]. We showed that all major wine cultivars were infected, often with multiple viruses. The most prevalent viruses detected in *V. vinifera* wine grape cultivars were GRSPaV, GLRaV-3, GPGV, and GRBV [4]. 

French-American hybrids and North American grape species (hereby collectively referred to as non-*vinifera* grapes for simplicity) constitute an important part of Ontario’s grape/wine industry. These locally important non-*vinifera* grapes had been widely planted in the province, though large acreages of these grapes have been gradually replaced by grape cultivars of *V. vinifera* in recent decades. According to the 2016 Annual Report of Grape Growers Ontario, hybrid wine grapes account for 43% of total grape production in the province.

The objective of this study was to assess the prevalence of viruses among non-*vinifera* grapes in Ontario. Using multiplex reverse transcription polymerase chain reaction (RT-PCR), we show that all non-*vinifera* grapes were infected with some of the major grapevine viruses that are targeted in grapevine certification programs in major grape-producing countries. Test results were confirmed through HTS using the Illumina platform. 

## 2. Materials and Methods

### 2.1. Sample Collection and Processing

A total of 533 grape leaf samples representing 2665 vines from 63 vineyard blocks of 18 vineyards were collected during late summer to early fall in 2016 and 2017 from Niagara Peninsula and Prince Edward County (Appendix A and Table 1). These samples covered hybrid wine grapes (Vidal, Baco, De Chaunac, Chambourcin, Marquette, Marechal Foch, Frontenac), juice grapes (Niagara and Concord) and the table grape Sovereign coronation (for simplicity, we refer it to Coronation) (Table 1). The sampling was done using a similar method to that used in the virus survey for vinifera wine grapes [4]. Briefly, six to twelve samples were randomly collected, based on the size of the block, from each cultivar/block, and each sample included a total of ten leaves, with two basal leaves from each of the five vines in a panel. All collected samples were ground into fine powder with a mortar and pestle in liquid nitrogen and stored in conical tubes in a −20 °C freezer for the isolation of nucleic acids.

### 2.2. Isolation of Total RNA

Total RNA was isolated from each of the 533 samples using a modified protocol that was developed in our lab based on the Spectrum Plant Total RNA Kit from Sigma (St. Louis, MO, USA) [12]. 

### 2.3. Primers and Multiplex RT-PCR 

A panel of 14 target viruses were tested with multiplex RT-PCR for the collected samples. These include five viruses associated with the leafroll disease complex (GLRaV-1, -2, -3, -4, and -7); three viruses associated with the rugose wood complex (GVA, GVB and GRSPaV); three viruses involved in infectious degeneration and decline (GFLV, ToRSV and ArMV); GFkV; and two recently discovered viruses: GRBV and GPGV (Appendix A). The same multiplex RT-PCR system, which was established earlier in our laboratory for a large-scale survey of viruses in *V. vinifera* grapes [4], was used in this study. Primers for each target virus were designed based on the consensus sequence of multiple genetic variants of that virus for which genomic sequences were available in GenBank. This was to ensure that a broad spectrum of genetic variants of each virus could be detected. Each primer pair was shown to be specific to only the target virus. The amplicons of the set of primers to be used in a multiplex RT-PCR differ in size such that they would be readily distinguished after gel electrophoresis of the PCR products (Appendix A). cDNA synthesis was carried out using the High-Capacity cDNA Reverse Transcription Kit (Life Technologies), essentially as described in Xiao et al. (2018) [4]. A total of 1000 ng of total RNA was used as a template and primed with random primers.

### 2.4. High-Throughput Sequencing

Based on the results of virus survey with multiplex RT-PCR, hybrid wine grapes (Vidal and Baco), juice grapes (Niagara and Concord), and the table grape Coronation were selected for metagenomics analysis with HTS. Total RNAs from either a single sample or combined RNA preps from two or three vines were first subjected to removal of ribosomal RNAs (rRNAs) by using the Ribo-Zero rRNA Removal Kit from Novogene (Sacramento, CA, USA). The composition of samples used for HTS was as follows: Coronation, ON595; Vidal, ON936 and ON1030; Baco, ON562 and ON1193; Niagara, ON544, ON919, and ON1206; and Concord, ON602, ON721, and ON766. These samples were collected from six individual grape growers located in Niagara, Ontario. 

The RNA samples after rRNA removal were used as templates to prepare a cDNA library using the Illumina TruSeq RNA Sample Prep Kit. Sequencing was carried out on an Illumina NovaSeq 6000 sequencer generating 150-bp pair-end reads at Novogene. The HTS data sets were first analyzed using CLC Genomics Workbench (Qiagen, Hilden, Germany). For de novo assembly, the raw sequencing reads were filtered to remove adaptor sequences and reads of low quality, and then mapped to the reference genome of *V. vinifera* (PRJEA18785) to eliminate host sequences. Non-grapevine sequence reads were then de novo assembled into contigs. De novo assembly was done by mapping reads back to contigs with the default parameter setting and a minimum contig length of 250nt. Resulting contigs were subsequently used as queries in a BLAST search against the complete reference sequences of viruses and viroids (http://www.ncbi.nlm.nih.gov/genome/viruses/ (accessed during 2018–2023 for several times)) to identify viruses and viroids that were present in the samples. The default threshold of 10 was used as the EXPECT value. To obtain the complete or partial genome sequences of the viruses and viral variants detected, de novo assembled viral contigs were compared manually with individual viral genome sequences available in GenBank. 

The raw sequence data from HTS were deposited in the NCBI SRA database under BioProject accession PRJNA1003946 and SRA accessions SRR25590732-SRR25590735. The complete and near-complete genome sequences of viruses identified through HTS were deposited in GenBank under the following accession numbers: OR478438-OR47843841 for GRSPaV; OR478442-OR478446 for GLRAV-3; OR478447 for GLRaV-2; OR478448-OR478450 for GRBV; OR400565-OR400566 for GVA; and OR400567-OR400568 for GVE.

## 3. Results

### 3.1. Status of Viral Infections among Non-Vinifera Grapes

The 533 samples collected from 63 non-vinifera vine blocks were screened using multiplex RT-PCR system for 14 viruses. Ten distinct viruses were identified in the sampled grapevines, specifically GLRaV-1, -2, and -3; GRSPaV; GVA; GVB; GRBV; GPGV; GFkV; and ToRSV; as illustrated in Figure 1. Their prevalence ranged from 65.2% for GRSPaV to 0.2% for GLRaV-1 (Figure 1). Among the identified viruses, the most common were GRSPaV, found in 65.2% of the samples, followed by GLRaV-3 at 52.9%, GPGV at 29.1%, GRBV at 15.3%, GVB at 13.4%, and GFkV at 11.8% (Figure 1). GLRaV-1, GLRaV-2, and ToRSV were detected only in hybrid wine grapes, but not in juice or table grapes (Table 2). GVA was detected in both juice and hybrid wine grapes, but not in table grapes, while GVB was detected in both table grapes and hybrid wine grapes, but not in juice grapes (Table 2). Overall, 83.7% of the samples tested positive for at least one virus (Figure 2). The survey results also revealed a high percentage of mixed infections, with 26% of the samples testing positive for two viruses, 19.1% for three viruses, 9.4% for four viruses, and 7.1% for five or more viruses (Figure 2).

Overall, GRSPaV, GLRaV-3, and GPGV were most common among non-*vinifera* grapes as they were detected in all 10 grape cultivars (Table 2). GFkV was detected in nine of the ten cultivars. It is interesting to note that GRBV was also very common among these non-*vinifera* grapes, except three hybrid wine cultivars (Marquette, Marechal Foch, and Frontenac). In contrast, three viruses had a very low prevalence among these non-*vinifera* grapes. For instance, GLRaV-1 was detected in a single Baco sample, ToRSV was detected in only three hybrid wine grape cultivars, and GLRaV-2 was only detected in 30.8% of samples of Vidal grapes (Table 2). 

Based on the results of RT-PCR, Vidal was the most severely infected among all the non-*vinifera* grapes tested, as it was infected with the largest number of viruses (9 viruses) and the highest infection rate, with GRSPaV (90.5%), GLRaV-3 (65.4%), GPGV (56.4%), GFkV (39.7), GVB (38.5%), GLRaV-2 (30.8%), and GRBV (11.5%) (Table 2). It is worth noting that all the samples tested positive for GLRaV-2 in this study were Vidal. This cultivar also had the highest infection rate for GVB (38.5%) compared with all other hybrid grape cultivars we surveyed. Baco ranked second in virus infection as it was infected with eight viruses, among which GRSPaV (55.4%), GRBV (45%), GVB (37.5%), GLRaV-3 (36.3%), and GPGV (26.3%) were predominant (Table 2). It is also important to note that Baco had the highest infection rate for GRBV (45%), followed by GVB (37.5%) among all the non-*vinifera* grapes tested. De Chaunac was infected with six viruses and the predominant ones were GRSPaV (72.7%), GLRaV-3 (50%), ToRSV (22.8%), and GPGV (13.6%) (Table 2). Chambourcin, Marquette, Marechal Foch, and Frontenac were all infected with five viruses each, and the types of viruses differed according to the cultivar (Table 2). For example, Chambourcin had the highest infection rate for GPGV (58.3%) among all the non-*vinifera* grapes tested, Marquette for GVA (25.0%), and Marechal Foch for GLRaV-3 (83.9%). Finally, Frontenac appeared to be the least infected: although five viruses were detected in Frontenac samples, the predominant virus was GRSPaV (32.7%) (Table 2), which is the most common among all types of grapevines regardless of their genotype or uses [13].

The two most common juice grape cultivars grown in Canada and the Northeastern United States, Concord and Niagara, were each infected with the same five viruses: GRSPaV, GLRaV-3, GPGV, GRBV, and GVA, with 10% of the Niagara samples also being positive for GFkV (Table 2). As for Concord samples, GLRaV-3 was the most prevalent (64.4%), followed by GRSPaV (40.8%), GRBV (23.3%), and GPGV (13.6%). As for Niagara samples, the most common virus was GRSPaV (73.3%), followed by GLRaV-3 (67.1%), GPGV (25.7%), GVA (18.6%), and GRBV (12.9%) (Table 2). Lastly, six viruses were detected in the table grape Coronation, with GRSPaV being the most prevalent (63%), followed by GLRaV-3 (44.4%), GPGV (7.4%), GRBV (13.0%), GFkV, and GVB (Table 2).

### 3.2. Metagenomic Analysis Reveals Viromes of Varying Levels of Complexity among Hybrid Wine and Juice Grapes 

To confirm the results of RT-PCR tests and to further explore the virome of these locally important non-*vinifera* wine grape cultivars, we conducted RNA-seq analyses of two major hybrid wine grape cultivars (Vidal, a popular white grape used for making the famous icewine and Baco, a dark-berried wine grape cultivar), two commonly grown juice grapes (Niagara and Concord), and the major table grape (Coronation). Results show that each of these grapes were infected with multiple viruses but with different levels of complexity. Below, we briefly describe the composition of the virome identified in each grape cultivar.

#### 3.2.1. Vidal

Among the 88,054,982 sequence reads generated by HTS, 3.84% (3,379,815 reads) did not match the reference *V. vinifera* genome. Mapping against the GenBank database of viruses and viroids revealed that 22.7% of these non-grapevine reads (765,538 reads) matched sequences of 14 viruses and 3 viroids (Table 3). The total read counts corresponding to viruses and viroids accounted for 0.87% of the total sequence reads. The virome of Vidal was very complex as it was composed of 14 viruses, some of which contained multiple genetic variants (Table 3 and Table 4). These viruses included GLRaV-2 and GLRaV-3 of the family *Closteroviridae*, six viruses of the family *Betaflexiviridae* (GRSPaV, GPGV, GVA, GVB, GVE, and GVG), and five viruses of the family *Tymoviridae* (GFkV, GSyV-1, GAMaV, GRVFV, and GRGV). For four of these viruses (GLRaV-2, GLRaV-3, GVE, and GSyV-1), two distant genetic variants each were detected in the Vidal samples. Importantly, while one of the GLRaV-3 variants matched isolate WA-MR with 97–100% sequence identity, the other variant differed significantly in nucleotide sequence from all GLRaV-3 isolates for which genome sequences were available at the time in GenBank. This variant whose complete genome sequence was subsequently obtained and deposited in GenBank under the accession number MK032068 was designated as Vdl [14]. Similarly, Vidal samples were also infected with two distinct variants of GVE, one of which potentially represents a new and distinct variant that is only 73–83% identical to isolate WAHH2 from Cabernet Sauvignon [15].

#### 3.2.2. Baco

In sharp contrast to Vidal, the virome of Baco is much simpler. Only four viruses were detected in the composite RNA sample representing two Baco samples (ON562 and ON1193) that were subjected to HTS analysis (Table 3): GRBV, GPGV, GRSPaV, and GVB. In addition, the read counts for all four viruses were much fewer than those from the Vidal dataset. For example, GRBV had the highest read count at 5881, whereas the next most abundant was GPGV, at 3248 reads (Table 3). Furthermore, the sequence reads matched a single genetic variant for each virus (Appendix A). We would like to note that the remaining two viruses all had significantly fewer sequence reads: GRSPaV at 539 reads and GVB at 110 reads (Table 3).

#### 3.2.3. Niagara

Among the composite of three Niagara samples that were pooled for RNA-Seq, six viruses were detected, which were GRSPaV (44.3%), GLRaV-3 (39%), GPGV (6.4%), GVE (6.4%), GRBV (2.5%), and GVA (12%) (Table 3). The sequence contigs assembled represented six genetic variants of GLRaV-3, three variants of GRSPaV, and a single variant each for the remaining viruses (Table 5). Interestingly, in addition to these six viruses that are commonly detected in *V. vinifera* cultivars, 744 sequence reads matched the three genomic segments of TSV isolate Illinois, though with different levels of genome coverage and sequence identities. For example, two contigs matched RNA 1 of TSV with 99% nt sequence identity and 66.9% genome coverage. Similarly, two contigs matched RNA 2 of the virus, with 100% sequence identity and 59.4% genome coverage. In contrast, a single contig matched RNA 3 with 97% genome coverage but only 91% identity.

#### 3.2.4. Concord

HTS analyses revealed four viruses in the composite samples containing three Concord samples: GRSPaV, GRBV, and GLRaV-3 (Table 3). Sequences related to GRSPaV are not only the most abundant but also the most diverse in the composite samples as they represent multiple genetic variants, albeit at various levels of genome coverage (Appendix A). All sequence reads of GLRaV-3 mapped to isolate WA-MR with 98–100% sequence identity. Similarly, all sequence contigs matching GRBV were identical to isolate 93–26. 

#### 3.2.5. Coronation

This table grape cultivar was the least infected in terms of the number of viruses detected and the second lowest in terms of the total count of sequence reads related to viruses among the samples analyzed by HTS (Appendix A). The total sequence count related to viruses is small, at 22,267. Three viruses were detected in this single Coronation sample, ON595, with GPGV being the predominant (21,178 reads), followed by TSV (793 reads) and GRSPaV (223 reads). 

## 4. Discussion

It is common that both *vinifera* and non-*vinifera* grapes are grown adjacent to one another or in close vicinity. As such, non-*vinifera* grapes could serve as reservoirs of viruses for infecting *V. vinifera* vineyards through transmission by vectors and vice versa. Earlier surveys conducted in Ontario [4] and in British Columbia [16,17] provided clear evidence that all major viruses were prevalent among vinifera wine grapes. However, only limited research was conducted toward the status of viral infection in non-*vinifera* grapes in the US [18,19,20,21], and virtually no information was available for Canada except a single survey at the time when this study was being carried out. MacKenzie et al. (1996) conducted the first and only national survey for four viruses (ArMV, GFLV, GLRaV-1, and GLRaV-3) among European grapes, as well as a small number of French-American hybrids and North American grapes of the *V. labrusca* origin [22]. Only six hybrid and four juice grapes collected from Ontario were included in this survey.

A province-wide survey for commonly targeted viral pathogens in commercial *V. vinifera* wine grape vineyards in Ontario were carried out in 2015 and 2016 and revealed that GLRaV-3, GRBV, and GPGV were the major viruses involved in the disease outbreaks [4]. As non-*vinifera* grapes (table grapes, juice grapes, and hybrid wine grapes) constitute an important part of Ontario’s grape/wine industry, it is necessary to understand their virus infection status. This is important because all types of grapes, regardless of their genetic background or uses, could serve as host to many of the viruses that are destructive to the grape and wine industry. Consequently, they would function as natural reservoirs for the spread of vector-transmitted viruses. Therefore, a large-scale, comprehensive survey was conducted in 2017–2018 to assess the distribution and prevalence of some viruses commonly targeted in non-*vinifera* grape vineyards in Ontario. We have tested for 14 viruses in 533 composite samples representing 2665 vines collected from major grape-growing regions in the province. We showed that virus infections in non-*vinifera* grapes were as prevalent as in vinifera wine grapes. Ten viruses were detected, and these viruses have varying degrees of prevalence, ranging from 0.2 to 65.2% (Figure 1). The four most prevalent viruses detected in non-*vinifera* grapes were GRSPaV, GLRaV-3, GPGV, and GRBV, although GRSPaV ranked first in terms of the percentage of samples infected while GLRaV-3 ranked first when considering the percentage of vineyard blocks that were infected (Figure 1). It is noteworthy that GRBV was detected in 15.3% of the samples collected from 26 of the 63 vine blocks (Figure 1). It is also interesting to note that GLRaV-1 has a very limited distribution among non-*vinifera* grapes, as a single Baco sample tested positive for the virus (Table 2). In the same period of this survey, Poojari et al. (2020) [23] conducted a similar survey targeting seven viruses (GLRaV-1, -2, -3, -4; GRBV; GPGV; and GFLV) in six hybrid cultivars and several vinifera grapes, including three hybrid cultivars that were also included in our survey—Vidal, Marechal Foch, and Marquette. These authors reported that GLRaV-3 was the most prevalent, followed by GPGV, GLRaV-1, and GRBV. GLRaV-2 and GLRaV-4 were not detected in the samples tested [23]. 

Data obtained from HTS analyses are in good agreement with those from RT-PCR and revealed different degrees of complexity of the viromes among five grape cultivars representing hybrid wine grapes, as well as table and juice grapes of *V. labrusca* origin. For example, 14 viruses were detected in Vidal (Table 4), a popular white grape that is predominantly used for making the world-famous icewine. Based on the counts of sequence reads, the most abundant viruses detected in Vidal samples were GRSPaV (24.7%), GLRaV-2 (24.6%), GVB (16.9%), and ArMV (13.5%), whereas sequence reads corresponding to GLRaV-3 account for only 6%. In a separate HTS analysis conducted by another group, Fall et al. (2020) [24] used dsRNA sequencing to determine the virome of samples from Vidal and Pinot noir. They found that the predominant viruses in the Vidal samples were GRSPaV, GLRaV-3, and GLRaV-2. In sharp contrast, Baco, a popular dark wine hybrid grape that is grown in North American and some countries of Europe, had a much less complex virome comprising only four viruses (Appendix A). 

On the other hand, the Niagara virome was also very complex. Collectively, RNA-seq data revealed seven viruses including GLRaV-3, GRBV, and four viruses of the *Betaflexiviridae* family (Table 5). Strikingly, the sequence contigs related to GLRaV-3 belong to five distinct genetic variants, including one that is more closely related to isolate Vdl from Vidal [14]. 

It was not expected that a considerable number of sequence reads closely related to TSV were detected through HTS from both Niagara (744 reads) and Coronation (793 reads). This is surprising as TSV has never been reported to infect grapevine. Though the sequence read counts of TSV in both samples are relatively low, the contigs that were generated from the assembly of these sequence reads had significant genome coverage when compared to the three genomic RNA segments of the isolate Illinois of TSV. For example, the Niagara contigs had between 59.4–97% coverage, while those from Coronation had 69.6–91.7% coverage (Table 5 and Appendix A). Given the considerable number of sequence reads and the extent of genome coverage, it is unlikely that the presence of TSV sequences in both grapevine cultivars was due to cross-contamination during the process of sequencing. However, the validity of this preliminary finding must be confirmed through further investigation.

It is puzzling as to why Vidal contains such a complex virome. Vidal is a complex hybrid wine grape developed in the 1930s by French breeder Jean-Louise Vidal as a potential cultivar for cognac production in western France [25]. In the late 1940s, this cultivar was brought to Canada by enologist De Chaunac. This is a winter-hardy cultivar and has been widely grown in the Niagara region of Ontario and the Okanagan Valley of British Columbia, with about 800 hectares of plantation. This cultivar accounts for two-thirds of the total production of hybrid wine grapes in Ontario, where it is mostly commonly used for icewine production [26]. It is also widely grown in several regions of the US, including the Finger Lakes in New York. Major contributing factors for the highly complex virome in Vidal may be the transmission of different viruses by biological vectors and the prolonged accumulation of these viruses over the long time span of old Vidal vineyards that were established many decades ago. To this end, GLRaV-3 and the four vitiviruses are known to be transmitted by mealybugs and scale insects [27], ArMV is transmitted by nematodes [28], and GPGV is transmissible by mites [6]. 

Viruses and their impact on non-*vinifera* grapes have not received much attention by the grape and wine industry or the grape virology community. Infections of non-*vinifera* grapevines with viruses usually cause no or only mild symptoms indistinguishable from those caused by physiological conditions such as nutrient deficiency. This study revealed that non-*vinifera* grapes are also infected with many viruses similar to vinifera wine grapes in Ontario vineyards. Therefore, it is necessary to investigate the impact of viral infections on the yield and quality of hybrid wine grapes. Given the unique history of grape growing in North America, where non-*vinifera* grapes have been gradually replaced by *V. vinifera* grapes in recent decades, both types of grapes are often grown in close proximity to each other. For the management of viral diseases in a vineyard or a region, effective disease control can be achieved only when mitigation strategies are applied to both *vinifera* and non-*vinifera* grape blocks. Furthermore, effective measures should also be employed to prevent specific viruses found only in non-*vinifera* grapes from spreading to vinifera grapes, in which they may adapt and cause severe diseases. Finally, the clean plant certification and registration program being developed in Canada must consider both *vinifera* wine grapes and non-*vinifera* grapes that are used for wine making, juice, or as table grapes. In summary, this study provides a framework for the grape and wine industry in developing strategies to manage grapevine viral diseases in countries where both types of grapes are grown.

## Figures and Tables

**Figure 1 viruses-15-01949-f001:**
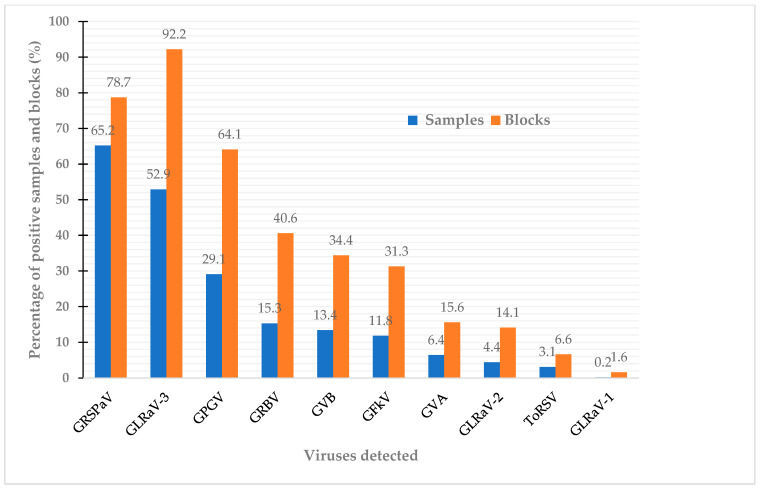
Prevalence of major viruses in juice, table, and hybrid wine grapes in Ontario based on RT-PCR. For the full names of viruses, refer to the list of abbreviations.

**Figure 2 viruses-15-01949-f002:**
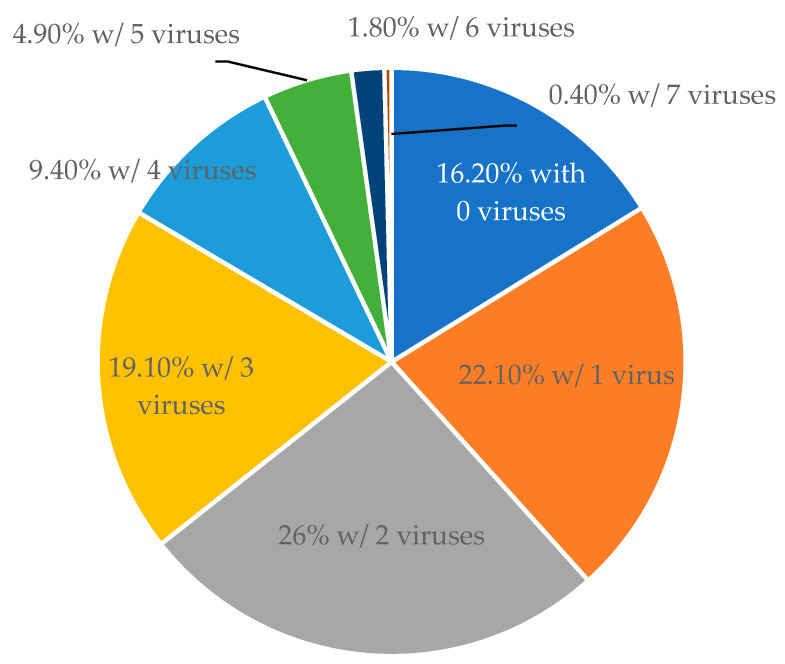
The percentage of grapevine samples that were infected with a single or multiple viruses. To ensure accuracy and consistency of data presentation, 488 samples that were tested for all 14 viruses were used to in the calculation.

**Table 1 viruses-15-01949-t001:** Sampling of three categories of non-*vinifera* grapes included in the survey.

Type of Grapes	Cultivar	No. of Vineyard Blocks	No. of Samples
French-American hybrid wine grapes	Vidal	10	78
	Baco	7	80
	De Chaunac	7	44
	Chambourcin	4	24
	Marquette	3	44
	Marechal Foch	5	31
	Frontenac ^1^	5	49
North American grapes	Concord	7	59
	Niagara	8	70
Table grapes	Sovereign coronation	7	54
Total	10	63	533

^1^ Includes Frontenac, Frontenac blanc, and Frontenac gris.

**Table 2 viruses-15-01949-t002:** Prevalence of major grape viruses in juice, table, and hybrid wine grapes in Ontario, Canada. Percentage of samples tested positive for each virus for each cultivar is shown. For the full names of viruses, refer to the list of abbreviations.

Cultivars	GRBV	GLRaV-1	GLRaV-2	GLRaV-3	GRSPaV	GPGV	GVA	GVB	GFkV	ToRSV
North American juice grapes:										
Niagara	12.9	0	0	67.1	73.3	25.7	18.6	0	10	0
Concord	23.3	0	0	64.4	40.8	13.6	1.7	0	0	0
North American table grapes:										
Coronation	13	0	0	44.4	63	7.4	0	5.6	37	0
French-American hybrid wine grapes:										
Vidal	11.5	0	30.8	65.4	90.5	56.4	7.7	38.5	39.7	4.8
Baco	45	1.25	0	36.3	55.4	26.3	5.0	37.5	2.5	0
De Chaunac	2.3	0	0	50	72.7	13.6	0	0	6.8	22.8
Chambourcin	29.1	0	0	17	83.3	58.3	0	0	4.2	0
Marquette	0	0	0	40.9	52.2	15.9	25.0	0	2.3	0
Marechal Foch	0	0	0	83.9	87.1	32.3	0	9.7	22.6	0
Frontenac	0	0	0	4.1	32.7	8.2	0	0	8.2	4.1

**Table 3 viruses-15-01949-t003:** Summary of sequence reads corresponding to different viruses in Vidal, Baco, Niagara, Concord, and Coronation grape samples derived from RNA-Seq. For the full names of viruses, refer to the list of abbreviations.

Names of Viruses and Viroids	Read Counts (% of Total Viral Reads)				
	Vidal	Baco	Niagara	Concord	Coronation
Total reads	88,054,982	80,463,420	92,320,068	100,548,656	94,379,964
Reads not matching grapevine sequence	3,379,815	1,780,046	5,425,877	2,615,202	2,163,617
GRBV	-	5881	6798 (2.5)	1551	-
GLRaV-2	186,764 (24.6)	-	-	-	-
GLRaV-3	45,254 (6.0)	-	105,966 (39)	1044	-
GRSPaV	187,726 (24.7)	539	120,285 (44.3)	46,423	223
GPGV	8913 (1.2)	3248	17,264 (6.4)	-	21,178
GVA	3963 (0.52)	-	3141 (1.2)	-	-
GVB	127,982 (16.9)	110	-	-	-
GVE	53,004 (7.0)	-	17,323 (6.4)	-	-
GVQ	2595 (0.34)	-	-	-	-
ArMV	102,627 (13.5)	-	-	-	-
GFkV	34,892 (4.6)	-	-	-	-
GSyV-1	2138 (0.28)	-	-	-	-
GAMaV	1764 (0.23)	-	-	-	-
GRVFV	542 (0.07)	-	-	-	-
GRGV	720 (0.095)	-	-	-	-
TSV	-	-	744 (0.27)	-	793
Total viral reads	758,856	9778	271,321	49,018	21,401
Viroids (HSVd, GYSVd1 and GYSVd2)	6626	4362	17,483	7003	26,455

**Table 4 viruses-15-01949-t004:** Summary of sequence contigs from Vidal samples and their mapping to viruses and viroids. For the full names of viruses, refer to the list of abbreviations.

Viruses	GenBank Accession No.	Reference Isolates	Sequence Identity (%)	No. of Contigs	Genome Coverage (%)
GLRaV-2	KX774192.1	ISA-BR	99	1	100
	FJ436234.1	OR1	99–100	3	88
GLRaV-3	GU983863.1	WA-MR	97–100	3	99
	MK032068.1	Vdl	100	1	99
GRSPaV	KX925556.1	TEMP-BR	98	1	88
	KX925556.1	TEMP-BR	94	1	73
	KX274275.1	SK704-B	94	1	96
	KX274275.1	SK704-B	96–98	11	94
	AY881627.1	BS	97	1	58
	KT948710.1	VF1	94	2	99
	HE591388.1	PG	97–98	7	99
	FR691076.1	MG	98	1	93
	KX958435.1	CS-BR	97–98	2	62
	AY368590.1	Syrah	93–96	2	31
GPGV	KR528581.1	Tannat-GvPGV	97–99	9	100
GVA	DQ855084.2	GTG11-1	82	1	99
GVB	KY426923,1	8415	92	13	80
GVE	JX402759.1	WAHH2	98	1	100
	JX402759.1	WAHH2	73–83	5	94
GVG	MF405923.1	VID561	68–78	3	28
ArMV	AY303786.1	NW (RNA1)	84	1	97
	AY017339.1	NW (RNA2)	89	1	78
GFkV	AJ309022.1	M48	89	10	78
GSyV-1	KT037017.1	MH	95–97	2	94
	KX130754.1	TRAJ-BR	96	1	61
GAMaV	KY123917.1	CS	86–91	3	28
GRVFV	KY513701.1	Mauzae	82–86	5	94
GRGV	KX171167.1	Graciano-T53	87–90	2	29

**Table 5 viruses-15-01949-t005:** Summary of sequence contigs from Niagara samples and their mapping to viruses and viroids. For the full names of viruses, refer to the list of abbreviations.

Viruses	GenBank Accession Numbers	Reference Isolates	Sequence Identity (%)	No. of Contigs	Genome Coverage (%)
GLRaV-3	GQ352632.1	623	99–100	2	98.8
	GU983863.1	WA-MR	100	1	98.6
	GU983863.1	WA-MR	93–96	6	86.5
	KY073324.1	8415B	100	1	99.7
	JQ655295.1	Vdl	91.9–93.5	3	97.6
GRSPaV	KR054734.1	JF	98	1	99.1
	FR691076.1	MG	98	1	100
	KX035004.1	SGM5 clone 1	98	2	99.3
GVA	KC962564.1	I327-5	84	1	99.9
GPGV	KM491305.1	MER	99	1	98
GVE	GU903012.1	SA94	76	1	97
TSV	FJ403375.1	Illinois, RNA1	99	2	66.9
	FJ403376.1	Illinois, RNA2	100	2	59.4
	FJ403377.1	Illinois, RNA3	91	1	97.0
GRBV	KY426922.1	93–26	100	1	100

## Data Availability

The data presented in this study are available in the article.

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
