# Peer review of "Molecular and Metagenomic Analyses Reveal High Prevalence and Complexity of Viral Infections in French-American Hybrids and North American Grapes"

_viruses, 2023, doi:10.3390/v15091949_

Round 1
Reviewer 1 Report
The present article is the the most comprehensive study of viruses among non-vinifera grapes. Authors evaluated the prevalence of viruses and the levels of distribution by a combinatorial approch based on RT-PCR assay followed by a HTS study. The study is totally missing of statistical analysis. Add it for the manuscript publication.
Minor editing of English language required.
Author Response
Dear reviewer:
Thank you for going through the manuscript and for providing the review comments.
We are unsure what part of the data would require statistical analysis as suggested by this reviewer. The purpose of this work was to probe the prevalence and distribution of major known viruses among non-Vitis vinifera grapes that are still grown in Canada and part of the USA. Was this reviewer referring to the detection of the three viruses (TSV, ToNSV, GaIV) based on the relatively low sequence read counts? We would like clarifications from this reviewer. If this is the case, this would be a very valid point. We removed the sequence reads related to grapevine-associated illarvirus (GaIV) and tomato necrotic ringspot virus from the HTS data as their read counts are simply too low. We have also removed the claim on the first detection of TSV in grapevine. However, because the read counts for TSV are considerably higher in both Niagara and Coronation samples, we decided to retain the data related to TSV in the manuscript but without making any definitive interpretations or claims. Since the work was done over five years ago, Dr. Huogen Xiao, the key researcher of the work had moved on to a different institution that is far away from Guelph where my lab is located. It would be a real challenge to perform follow-up work in order to validate the presence of TSV in the grapevine samples. For this reason, we decided to remove such a bold and presumptive claim from the manuscript. We also included a paragraph in the Discussion to voice the limitations of our findings and to stress the importance of further research to validate our findings.
We hope you will find the modifications we made to the manuscript acceptable. I also attach a document containing all responses we have made to each review comments raised by all three reviewers, in case you would like to see them.
Thank you again for taking the time to review the revision of our submission.
Best regards
Baozhong Meng

Reviewer 2 Report
In this study, the authors examined viral and viroid infections in hybrid grapes in Canada using RT-PCR (reverse transcription polymerase chain reaction) and HTS (high-throughput sequencing). The overall purpose of the study is suitable for publication, and the authors conducted intensive analyses. However, it would be beneficial to include information about the raw data from HTS and the viral genome sequences in the manuscript. Additionally, the manuscript's quality would be enhanced by including phylogenetic trees for representative viruses. Therefore, I recommend that the manuscript undergo major revision. Below are my specific comments:
L10-15: This introduction paragraph can be shortened.
L13: Write "Vitis labrusca" in italics.
L18: Spell out "Reverse Transcription Polymerase Chain Reaction (RT-PCR)."
L19: Spell out the names of the viruses.
L23: Spell out "High-Throughput Sequencing (HTS)."
L27: Provide the abbreviation for "Tobacco streak virus (TSV)."
The abstract should be less than 200 words. It can be shortened by focusing on the essential results from this study.
L47: Write "V. labrusca" in italics.
L71-84: Avoid using italics for 'Grapevine leafroll-associated virus 1' and other virus names according to ICTV instruction. Instead, write the virus names in normal font. For example, use "grapevine leafroll-associated virus 1 (GLRaV-1)" in normal font. Change the virus names accordingly.
L96, 109: Write "V. vinifera" in italics.
L131: Spell out "Reverse Transcription Polymerase Chain Reaction (RT-PCR)."
In the Materials and Methods, consider providing a geographical map showing the sampling regions in Canada for readers who are unfamiliar with the area.
L147: Check the degree sign.
L153: Provide the city and country names for Sigma.
L159-165: Although the authors referred to a previous study, it would be desirable to provide a brief description of RT-PCR.
L169: Replace "high-throughput sequencing" with "HTS." After introducing the abbreviation, use "HTS" throughout the text.
L172-173: Describe how the authors removed ribosomal RNAs and provide the name of the kit used.
L173: Use "HTS" instead of the full name.
L177: Provide the city and country name for Qiagen.
L178: Provide the threshold for the BLAST search.
L181: Please describe de novo assembly briefly.
L184: The raw data derived from HTS should be deposited in the SRA database in NCBI, and the accession numbers should be provided in the manuscript.
For the legend of Figure 1, include that viruses were detected by RT-PCR.
The pie chart in Figure 2 needs clarification. Revise the legend and figure to ensure readers can understand the results correctly. Additionally, consider coloring the characters for "0 viruses" (16.20%) in white if the background is blue.
L260: Provide a more detailed explanation of the samples, including the number of samples used for HTS and their geographical regions.
L262: Write "V. vinifera" in italics.
Table 3 should be positioned between sections 3.2 and 3.2.1.
Consider adding phylogenetic trees using the obtained complete or nearly complete genome sequences of identified viruses and viroids in the results.
Furthermore, the complete or nearly complete viral genome sequences covering the whole open reading frames (ORFs) should be deposited in GenBank, and the accession numbers should be provided in the manuscript.
Additionally, the raw data from HTS should be deposited in the SRA database in NCBI, and the accession numbers should be included in the manuscript.
Author Response
Thank you for the thorough review, detecting many of the mistakes, and more importantly for offering many excellent insights and suggested changes. Below we provide a point-by-point response to each of your comments. We provided a detailed and point-by-point response in a separate document entitled "Item by item response to review comments" that is submitted together with the revised manuscript.
Thank you for the help!

Reviewer 3 Report
Manuscript viruses-2531532 describes surveys of viruses in American-French hybrid grape cultivars by RT-PCR and high-throughput sequencing. The study is interesting because little information is available on the occurrence of viruses in American-French hybrid grape cultivars. However, essential information is missing in Materials and Methods, HTS seems to be been ran sub-optimally, some conclusions do not seem to be justified by the data, and the manuscript is overall not well written; it is verbose and many paragraphs and sentences are superfluous. See specific comments below:
Line 2 and throughout the manuscript: What is a heavy viral infection? Heavy has no scientific merit. Do you mean high infection rate? Or, high virus prevalence?
Line 3 and throughout the manuscript: vinifera should be italicized
Lines 10-12: Eliminate these sentences that are unrelated to the research presented in the manuscript.
Line 13 and throughout the manuscript: Italicize Vitis labrusca
Line 13: Not every cultivar used in this study derives from Vitis labrusca. Some hybrids result from crosses with V. riparia or V. rupestris.
Line 14: Change situation with occurrence
Line 14 and 15: ... viral diseases in hybrid wine, juice and table grapes remains ...
Lines 17-18: ... two juice grape cultivars, and the table grape cultivar 'Sovereign coronation'. Based ...
Line 18: eliminate but
Line 19: ... and GRBV were frequently detected. As ...
Lines 20 and 21: Change are to were
Line 23: ... grape cultivars, high-throughput sequencing (HTS) of five composite ...
Line 24: ... grapevine cultivars was performed. Results from HTS agreed with ...
Line 27-28: this is a bold statement First, it ignores the literature; see https://pubmed.ncbi.nlm.nih.gov/30831890/. More importantly, how can it be claimed that a new virus is detected with low HTS read counts (~700) and no positive and negative controls used in HTS. This goes without mentioning that the so called identification of tobacco streak virus is not validated beyond HTS. It would be nice to have ran RT-PCR to confirm the presence of tobaccos streak virus in 'Niagara' and 'Sovereign coronation'. Similarly how can the identification of tomato necrotic spot virus or grapevine-associated ilarvirus be claimed with even lower HTS read counts (200 and less, and 28, respectively)? These serious limitations of HTS and HTS data interpretation should be addressed.
Line 37: ... to the Mediterranean region and Central ...
Line 38: ... and powdery mildew, were ...
Line 39: ... destroying most vineyards. French ...
Lines 50-64: Eliminate this paragraph.
Lines 71: Eliminate 'they cause'
Line 71-83: Virus names should not be italicized or capitalized
Lines 88-97: Eliminate this paragraph
Line 104: Eliminate and
Line 109 and throughout the manuscript: Italicize V. vinifera
Lines 112-122: these sentences belnig to the Discussion
Line 122-129: Eliminate these sentences
Lines 137-148: Were the vineyards of hybrids selected for this study own-rooted or grafted onto a rootstock? This information could be essential to explain some of the multiple virus infections, as well as similarities acress cultivars and vineyard blocks.
Line 140 and throughout the manuscript; Change Chambercin to Chambourcin
Table 1: change Tables to Table in the first column
Line 175: Eliminate the
Figure 2: Vidal is describes with 9 viruses. Why are 9 viruses not included in this figure?
Line 211: Eliminate Closteroviridae and Betaflexiviridae from the header
Lines 182-184: Eliminate this sentence
Line 189: Eliminate except
Line 191: Change are to were
Line 192: ... and ToRSV. Their prevalence ...
Line 197: ... but not in table grapes ...
Figure 1: Change postive to positive
Line 225: ... were Vidal grapes. this cultivar had ...
Line 226: ... compared with all other ...
Line 247: ... table grape Coronation ...
Line 266: Eliminate extremely
Line 268: ... six viruses of the family Betaflexiviridae ...
Line 269: ... GVG, and five viruses of the family Tymoviridae ...
Line 270: which four viruses?
Line 273: Change were to are
Line 274: Eliminate at the time
Lines 274-275: This variant whose complete genome sequence was subsequently obtained and deposited in GenBank under the accession number MK0320068 was designed as Vdl [21].
Table 3 should be revised based on the above comments on the limitations of the HTS experiments.
Line 305: HTS analyses revealed six viruses in the three Concord samples: GRPaV, GRBV, GLRaV-3, GVE, GVB and GaIV (table 3). Sequences ...
Lines 325-326: ... understand their virus status. this is important ...
Lines 327: Change would to could
Line 329: ... spread of vector-transmitted viruses. therefore, ...
Line 331: ... some viruses in non-vinifera grape ...
Line 332: ... 14 viruses in 543 compositve samples ...
Line 336: Change are to were
Line 345: ... Marquette. these authors reported ...
Line 357-360: Eliminate this sentence
Line 361: Elimiante still
Line 365: Italicize Betaflexiviridae
Line 367-369: HTS data need to be validate, for instance, by RT-PCR before making this claim.
Lines 373 and 276: Change variety to cultivar
Lines 375: Changes 200 acres into hectares
Author Response
Dear reviewer:
We would like to thank you for the time and effort you made to thoroughly review our manuscript. We found the comments and suggested changes are valuable as they will certainly enhance the quality of our work.
We have addressed each and every comments you raised. Please see the attached document entitled "Item by item response to reviewer comments".
We welcome any further suggestions and comments you may have. Thank you for going through the revised version of the submission.
Best regards
Baozhong Meng

Round 2
Reviewer 2 Report
I am thoroughly satisfied with the revisions made.
I wholeheartedly recommend the manuscript for publication in its current form.
As the authors have pointed out, following the rules proposed by ICTV might lead to confusion.
In my perspective, adhering to the rule is advisable.
Please remember to write virus names in lowercase and abbreviate them accordingly.
For other instances, italicize scientific names, genera, and family names.
Author Response
Dear reviewer:
As the corresponding author and a colleague in the plant virology field, I fully respect your suggestions and, as a matter of fact, I am impressed with both the thoroughness and the quality of your comments and suggested changes. I express my gratitude and my high respect for you as a colleague and a plant virologist. Thank you for all you have done with the review of our manuscript!
Best wishes to you and your research!
Sincerely,
Baozhong Meng

Reviewer 3 Report
Revisions of improve the quality of the study. Nonetheless, additional editorial changes should be considered for further improvement. See recommendations below. tTe major issue with this research is the poor implementation of HTS with no positive and negative controls. This is a major flaw of the study. This is more so since conclusions are reached on the presence of some viruses based on low sequence read counts (<800) without any follow-up validation, for instance, by RT-PCR. There is an abundance of papers in the literature that report the occurrence of plant viruses based on the application of HTS with no validation of HTS data to only find out that a lot of these finds actually correspond technical artifacts. There is no need to add this type of misleading papers to the literature. Therefore, the so-called identification of viruses based on low read counts (<800) or extremely low read counts (<100) should be eliminated from this manuscript unless confirmed by RT-PCR.
L16: Eliminate 'with varying levels of distribution'.
L20: ... and a higher prevalence of individual viruses ...
L23: Change agree to agreed
L25: Change 'is host to' to hosted
L34: Change 'come from' to are
L54: Change Grapevine to grapevine
L94-95: Eliminate this sentence. It is not informative and referred to a figure of a previously published paper is not helpful.
L186: Eliminate 'as presented here'
L222-225: Eliminate all the 'at' in parenthesis
L239, Table 3: Extremely low read counts (30-230) and low read counts (500-800) should be eliminated. This is because no internal positive and negative controls were used in HTS; therefore, it is impossible to confidently assess how valid these low sequence read counts are.
L268: Change sequences to sequence
L277-283: TSV read counts are very low and the occurrence of TSV in Niagara was not validated in follow-up RT-PCR. Therefore, all information related to TSV should be deleted from this manuscript.
L279: What does TSV stand for?
L284: TSV information should be elimniated from table 5
L293-296: The read counts of GVB and GPGV are too low to be considered for ascertaining the occurrence of these two viruses in Concord. The corresponding information should be eliminated.
L303: Low read counts should be discounted
Line 362-373: Eliminate this paragraph
L403-407: Eliminate these two sentences
The quality of the English language is acceptable
Author Response
Dear reviewer:
We would like to take this opportunity to express our sincere gratitude to you for the very thorough and in-depth review of both versions of the manuscript. We found most of your comments and suggested changes are valid and helpful. Thank you.
We do recognize the limitations of HTS approach and the precarious nature of reaching conclusions solely based on HTS data. To ensure the reliability of the findings from HTS, we have checked the RT-PCR results for these samples that were used in HTS.
We removed all viruses with low read counts that are below 100. We hope you are satisfied with the revisions we made. For specifics, please see the attached response letter.
Thank you very much again for the time and effort you made in reviewing our manuscript and for the large numbers of very valuable comments.
Sincerely
Baozhong Meng
